# Effect of Cr on a Ni-Catalyst Supported on Sibunite in Bicyclohexyl Dehydrogenation in Hydrogen Storage Application

**Leonid M. Kustov** [1,2,3,*] **and Alexander N. Kalenchuk** [1,2]

1 Chemistry Department, Moscow State University, Leninskie Gory 1, Bldg. 3, 119992 Moscow, Russia
2 N.D. Zelinsky Institute of Organic Chemistry, Russian Academy of Sciences, Leninsky Prospect 47, 119991 Moscow, Russia
3 Laboratory of Nanochemistry and Ecology, National University of Science and Technology MISiS, Leninsky Prospect 4, 119991 Moscow, Russia
* Correspondence: lmk@ioc.ac.ru

**Abstract:** A comparison of the activity of mono- and bimetallic Ni-Cr/C catalysts deposited on a carbon carrier (sibunite) in the bicyclohexyl dehydrogenation reaction as a stage of hydrogen evolution in hydrogen storage systems is carried out. The interaction of Ni and Cr supported onto the carbon carrier—sibunite in bimetallic NiCr systems affects the change in the parameters of the crystal lattice of nickel, compared with the FCC lattice of Ni, as shown by the methods of XPS, TPR, XRD, high-resolution TEM and electron diffraction.

**Keywords:** hydrogen storage; dehydrogenation; bicyclohexyl; Pt-Ni-Cr supported catalysts; Ni-Cr alloys

## 1. Introduction

Recently, catalysis has become an increasingly important tool aimed at creating modern materials and technologies. For example, the successful replacement of lithium-ion batteries with cheaper methanol fuel cells largely depends on the development of new catalysts for the electrooxidation of methanol. It is known that the most active in this reaction, as well as in many other reactions, are supported systems based on noble metals, especially, platinum [1–3]. At the same time, these metals are prone to rapid CO poisoning in the methanol medium. The problem is solved by developing bi- and trimetallic catalysts, which not only increase resistance to the deactivating action of CO, but also allow one to reach a better performance of catalysts with a lower loading of precious metals. To increase the catalytic activity, transition metals (Fe, Ni, Co, Ag and others) are usually used as promoting additives [4–7]. The second group consists of metals that improve the corrosion stability of the catalysts (Cr, V, Mn) [8]. In particular, an increase in the conversion and selectivity in the methanol electrooxidation reaction with reduction of production costs is reported for bi- and trimetallic catalysts, such as PtCo, PtNi, PtFe, PtPb [9,10], PtNiCr and PtCoCr [11–13]. At the same time, the greatest resistance to CO poisoning is achieved when platinum is doped with ruthenium [14,15], including the addition (dilution) of third-base metals in the case of trimetallic systems, such as PtRuFe [16], PtRuNi [17], PtRuCo [18], PtRuW [19] and PtRuMo [20].

An increase in the activity of PtNiCr trimetallic catalytic systems supported on the carbon carrier, sibunite, was also found in [21] compared with mono- and binary systems applied in the bicyclohexyl dehydrogenation reaction. The reaction is crucial for hydrogen storage and hydrogen release systems based on the use of liquid organic hydrogen carriers (LOHC) as an alternative fuel system without the so-called carbon footprint instead of traditional hydrocarbon raw materials, since hydrogen oxidation does not produce $CO_x$ gases that pollute the atmosphere. In this reaction, the high activity of PtNiCr/C trimetallic

catalysts, as in the methanol electrooxidation reaction, is presumably achieved due to the stabilizing effect of the NiCr binary system on platinum [7]. In relation to hydrogen storage systems, the development of catalysts is limited by the need to carry out conjugated hydrogenation–dehydrogenation reactions without the formation of by-products of hydrogenolysis and cracking, which destroy the substrate and reduce the lifetime of hydrogen storage and release systems as a whole. This limits the choice of carriers and active metals; therefore, for the purposes of hydrogen storage, the study of the stimulating effect of promoting additives on the activity of Pt catalysts is of great interest. The aim of this study is to study the effect of the interaction of modifying base metals (Cr, Ni) on the activity of platinum in the bicyclohexyl dehydrogenation reaction.

## 2. Results

The morphology, average size of Ni and Cr particles and their distribution in bimetallic systems deposited on oxidized sibunite (C) and consisting of a mixture of Ni (3 wt.%) and Cr (1.5 wt.%) were studied using transmission electron microscopy (TEM). Figure 1 and Figure S1 (Supplementary Materials, SM) show an example of a micrograph of a catalyst (3Ni-1.5Cr)/C in a light (a) and dark field (b). It can be seen from Figure 1 that small particles of a regular shape are located on top of some large particles on the surface of the carbon carrier. The average size of large particles is ~20–30 nm and that of smaller ones is ~3–5 nm. The EDX data for this system, together with the analysis of interplane distances by the high-resolution TEM, show that the particles contain atoms of Ni and Cr metals coated with an oxide shell (NiO, $Cr_2O_3$). At the same time, a comparison of micrographs of TEM of a number of bimetallic Ni-Cr/C catalysts with their monometallic analogues showed that large particles belong to nickel and smaller ones are formed by chromium.

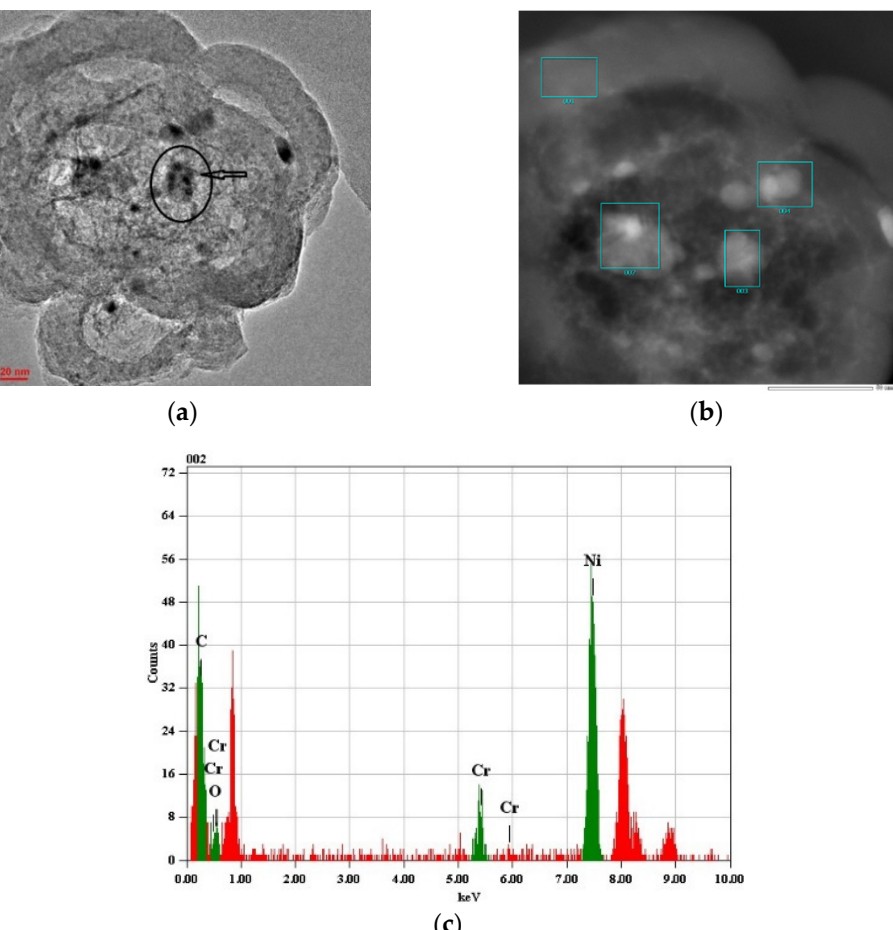

**Figure 1.** Catalyst (3Ni-1.5Cr)/C: micrograph in light (**a**) and dark field (**b**) with EDX (**c**).

Comparison with other investigated Ni-Cr systems indicates that the order of the application of metals does not have a noticeable effect on the change in the average size and distribution of particles on the surface of the sibunite. At the same time, the presence of $Cr_xNi_{1-x}$ solid solutions with different Cr/Ni ratios, as well as small amounts of $NiCrO_4$ with a rutile structure and $NiCr_2O_4$ with a spinel structure, is determined at the points of contact of the two metals at interplanar distances. Interestingly, the formation of the latter, together with a decrease in the degree of oxidation of the catalyst surface due to the alloying of nickel with chromium, is indicated as a possible reason for an increase in the activity of the PtNiCr system in the electrooxidation of methanol compared with a monometallic Pt catalyst [7,8]. It was shown [11–13] that the substitution of chromium for Mn and Co also affects the activity of the Ni-Mn and Ni-Co binary systems; however, the effect is less pronounced.

In order to identify the forms of nickel and chromium in the catalysts under study, the corresponding samples were examined using X-ray photoelectron spectroscopy (XPS). The comparison showed that the XPS spectra of all the analyzed systems, regardless of the order of application of metals, are identical and have no fundamental differences among themselves. In particular, only a doublet of broad lines with a poorly expressed structure and with a binding energy of the Cr $2p_{3/2}$ component of about 577.0 eV, which is typical for trivalent chromium compounds, is observed in the Cr 2p spectra of all the samples studied [22]. In the Ni 2p spectra of the studied samples, a small shoulder with a binding energy of about 852.7 eV is observed, which indicates the presence of metallic nickel [23]. At the same time, the main contribution to the Ni 2p spectrum is made by the oxidized state of nickel, which corresponds to the Ni $2p_{3/2}$ component with a binding energy of about 855.8 eV and an intense shake-up satellite, which is characteristic of divalent nickel [23]. The result of deconvolution of the spectra by the example of the catalyst (3Ni-3Cr)/C is shown in Figure 2 and in Table 1. The proportions of nickel in various states in this catalyst are indicated in comparison with the most characteristic data for other studied Ni systems.

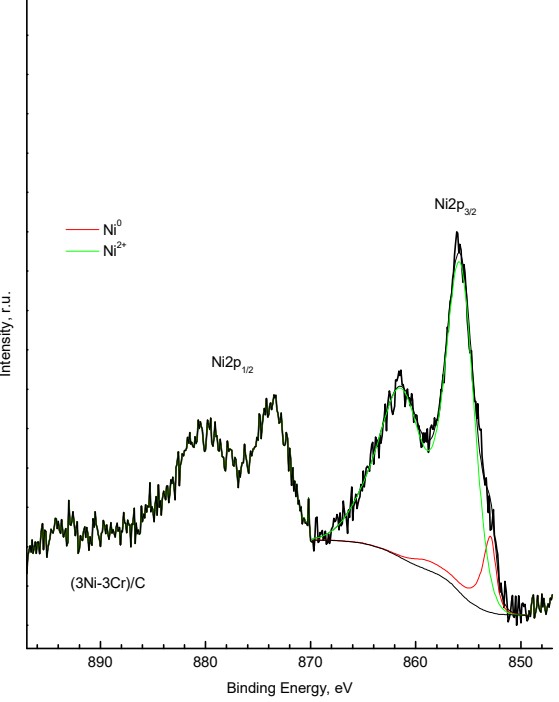

**Figure 2.** X-ray photoelectron spectra of Ni 2p electrons for (3Ni-3Cr)/C catalyst.

**Table 1.** The content of nickel in various degrees of oxidation on the surface of the studied samples according to the XPS data.

| Catalyst | Ni 2p | | | |
|---|---|---|---|---|
| | $E_b$, eV | Content, at.% | $E_b$, eV | Content, at.% |
| 3Ni/C | 852.5 | 53 | 855.8 | 47 |
| (3Ni-1.5Cr)/C | 852.5 | 27 | 855.4 | 73 |
| (3Ni-3Cr)/C | 852.7 | 8 | 855.4 | 92 |
| 1.5Cr/3Ni/C | 852.7 | 18 | 854.8 | 82 |

A comparison of the data in Table 1 shows that, regardless of the order of application of metals in the presence of chromium, the content of non-oxidized nickel (Ni(0)) in the studied bimetallic NiCr/C systems compared to the monometallic catalyst 3Ni/C decreases and the more strongly, the higher the Cr content. At the same time, it can be seen that there is a peak of Ni $2p_{3/2}$ electrons with binding energies $E_b$ = 852.7 eV in the spectra of catalysts with a low $Ni^0$ content, which is slightly higher than in the spectrum of 3Ni/C (852.5 eV) This may indicate that metallic nickel acquires some positive charge in these samples. This is also evidenced by the binding energy characterizing the second state of nickel in NiCr/C systems ($Ni^{2+}$). The value of the $E_b$ for this peak in the spectrum of the sample of 3Ni/C is equal to 855.8 eV, which is higher than that for the samples containing chromium. The difference in the electronic state of the components seems to indicate the presence of an interaction between these two metals. The effect is most pronounced for the 1.5Cr/3Ni/C catalyst, in which chromium is deposited on top of nickel.

The shape of the TPR profiles for the studied NiCr systems also shows a noticeable difference with individual monometallic catalysts. The TPR curves for the monometallic catalyst 3Ni/C [24] and 1.5Cr/C [25] demonstrate a number of characteristic peaks associated with the reduction processes of nickel and chromium. In contrast, in the NiCr systems, several processes occurring in a wide temperature range (T = 300–800 °C) are superimposed on the TPR profiles. This shape of the curves may be due to a variety of topochemical reactions responsible for the reduction of mixed Ni-Cr oxides, including the interaction between nickel and chromium noticed in the analysis of XPS spectra. At the same time, a low-temperature peak is highlighted in Figure 3, the position of which on the TPR curve for NiCr systems with different metal deposition, the order differs and is T = 138 °C for the catalyst (3Ni-1.5Cr)/C, T = 146 °C for 1.5Cr/3Ni/C and T = 170 °C for 3Ni/1.5Cr/C.

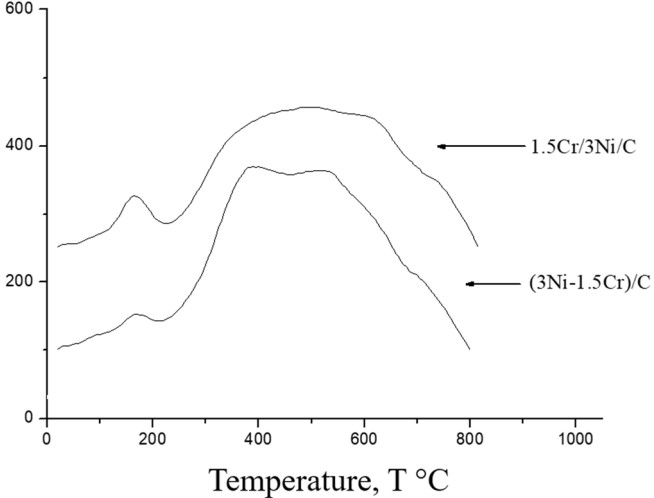

**Figure 3.** TPR profiles for bimetallic catalysts (3Ni-1.5Cr)/C and 1.5Cr/3Ni/C.

It is noteworthy that the position of each of these peaks decreases in comparison with both the Ni/C catalyst (T = 190 °C) and Cr/C catalyst (T = 265 °C) [25]. The intensity of the peaks for the bimetallic systems significantly exceeds those for monometallic catalysts. The authors [26,27] associated the increase in the intensity of TPR peaks with a decrease in the fraction of amorphous particles of metal oxides, including those produced due to the formation of the Ni-Cr alloy or bimetallic particles. It was also established that chromium doping of the Ni/MgO-La$_2$O$_3$ system leads to the modification of a part of the catalyst sites, with the formation of alloys with nickel and possibly the reduction of the size of crystallites [28]. Similar results were obtained for the Cu-Ni/Al$_2$O$_3$ system, where the addition of copper contributed to an increase in the stability of amorphous nanoalloy catalysts and a decrease in the Ni reduction temperature [29].

Comparison of monometallic Ni/C and bimetallic NiCr/C systems by the XRD method shows that the diffractograms of all the studied catalysts have peaks at 2θ = 25.7 and 43.4°, corresponding to the reflections of (002) and (101) of the carbon sibunite carrier (Figure 4). The maxima at 2θ = 44.4 and 51.8° are attributed to the crystallographic planes (111) and (200) of the FCC lattice of Ni (2θ = 44.505 and 51.86°; PDF 4-850). At the same time, the maximum of the (200) peak on the diffractogram of the monometallic catalyst 3Ni/C is shifted towards larger angles (2θ = 52.0°), as compared with bimetallic catalysts. The value of this shift corresponds to the lattice period a = 0.3514 (+0.0003) nm, which is lower than the table value for the FCC lattice of Ni (a = 0.3524 nm; PDF 4-850). Such an increase was associated [30] with the formation of metastable Ni$_3$C carbide, the enthalpy of formation of which is $\Delta H_{Ni3C}$ = 75.3kJ/mol. The enthalpy of formation of chromium carbides have large negative values ($\Delta H_{Cr3C2}$ = −9.5 kJ/mol; $\Delta H_{Cr7C3}$ = −164.4 kJ/mol; $\Delta H_{Cr23C6}$ = −341.4 kJ/mol); therefore, Cr atoms interact more actively with Ni than with the carrier when added to the bimetallic system. The Ni lattice periods calculated by two reflexes (111) and (200) in bimetallic NiCr systems exceed those for both the FCC lattice of Ni and the monometallic catalyst 3Ni/C [21]. At the same time, the most intense reflex 110 of the Cr BCC lattice (2θ = 44.37°, PDF 6-694) overlaps with the indicated reflex (111) of the Ni FCC lattice and does not appear on diffractograms, apparently due to the low content of metallic Cr in the volume of the catalyst or its X-ray amorphous state. In the absence of the contribution of chromium to the position of peaks on the graphs, this excess indicates the formation of Cr-Ni solid solutions formed by the type of substitution of nickel atoms for chromium. The broadening of the (200) Ni lines on the diffractogram indicates that the CrNi-solid solution has a heterogeneous character, in which there are regions with different chromium contents (Cr$_x$Ni$_{1-x}$).

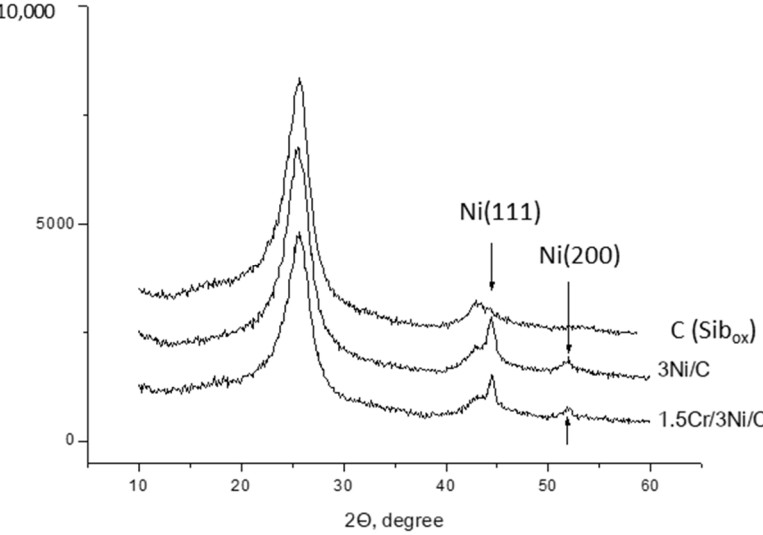

**Figure 4.** Diffractograms of the carbon carrier, 3Ni/C and 1.5Cr/3Ni/C catalysts.

The estimate of interplane distances of different sites obtained by the high-resolution TEM of the surface of the studied NiCr catalysts closely coincides with the data of the electron diffraction method. Two series of reflexes (111) and (200) are observed on electronograms of the NiCr catalysts: nickel (d(111) = 0.209 ± 0.006 nm and d(200) = 0.179 ± 0.002 nm) and chromium solid solution in nickel (d(111) = 0.215 ± 0.010 nm and d(200) = 0.187 ± 0.010 nm). The value d(111) of nickel in a two-component catalyst (3Ni-1.5Cr)/C calculated from the distance between adjacent parallel atomic planes is d = 0.2124 nm, which also exceeds the value of d(111) for the catalyst containing no chromium (d = 0.2091 nm). In accordance with the calculated parameters of the FCC lattice, the crystallographic structure of Ni-Cr is shown in Figure 5.

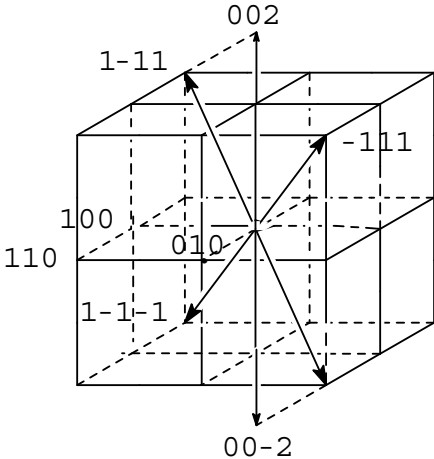

**Figure 5.** Crystallographic structure of the Ni-Cr phase.

The data obtained in studying the catalytic activity of catalysts in bicyclohexyl dehydrogenation showed that monometallic Ni and Cr catalysts exhibited a low activity in the selectivity formation of the final product, biphenyl ($C_{12}H_{10}$); thus, the rate of hydrogen evolution is low (Table 2). Both monometallic and bimetallic catalysts demonstrate a stable rate of hydrogen evolution and selectivities to the semi-hydrogenated and fully dehydrogenated products indicated in Table 2, without any noticeable loss for at least 8 h of operation.

**Table 2.** Catalytic properties of Ni-Cr/C catalysts of bicyclohexyl dehydrogenation (T = 320 °C, *p* = 1 atm, WHSV = 2.8 $h^{-1}$).

|  | 3Ni/C | 1.5Cr/C | 3Ni/1.5Cr/C | (3Ni-1.5Cr)/C | 1.5Cr/3Ni/C |
|---|---|---|---|---|---|
| Rate of hydrogen evolution, mol/h × $10^{-3}$ | 1.863 | 2.760 | 1.021 | 1.542 | 1.014 |
| $S_{C12H10}$, % * | 35 | 60 | 48 | 49 | 47 |
| $S_{C12H16}$, % * | 65 | 40 | 52 | 51 | 53 |

* Selectivities to the semi-hydrogenated and fully dehydrogenated products.

It is seen from Table 2 that the rate of hydrogen evolution in the course of conversion of bicyclohexyl on the Cr catalysts was noticeably higher than on the Ni catalysts; in turn, it is much lower than that observed on the Pt catalysts [21]. However, the selectivity for the final product, biphenyl ($C_{12}H_{10}$), on the Cr catalysts was noticeably higher than on the Ni catalysts. In the case of nickel, the formation of trace amounts of cracking products [24] was also observed. Since the content of cracking products ranged from 50 to 200 ppm, their contribution was not taken into account when calculating the selectivity of the main reaction products. No by-products were found for the studied bimetallic NiCr systems, which is important for hydrogen storage systems. According to chromato-mass spectrometry, in all the cases, the reaction mixture in addition to fully unsaturated biphenyl, contained one intermediate cyclohexylbenzene. At the same time, the rate of bicyclohexyl conversion

(hydrogen release) on NiCr systems, regardless of the order of introduction of metals, was even lower than on monometallic catalysts. This decrease correlates with a decrease in the content of non-oxidized nickel (see Table 1).

## 3. Materials and Methods

### 3.1. Preparation of Catalysts

Oxidized sibunite was used as a carrier (trademark C, Omsk, Russia), with an average granule diameter of 1.5–1.8 mm, a specific surface area of 243 $m^2/g$, an average pore size of 4.2 nm and a pore volume of 0.45 $cm^3/g$ [31]. Monometallic Ni and Cr catalysts deposited on a carbon carrier were prepared by incipient wetness impregnation with aqueous solutions of the corresponding salts, nitrates $Ni(NO_3)_2$ $6H_2O$ (chemical grade purity; Khimmed, Russia) and $Cr(NO_3)_3$ $9H_2O$ (chemical grade purity; "Acros Organics"), and bimetallic catalysts—Ni-Cr/C—by impregnation of Ni/C and Cr/C or joint impregnation of the sibunite. After impregnation, the samples were successively dried for 24 h in air at room temperature and then, for 4 h at a temperature of 130 °C. Further, the samples were calcined at a temperature of 500 °C in a current of $N_2$ (99.9%, 50 mL/min). Pt-containing catalysts were prepared by incipient wetness impregnation of the carrier or Ni/C, Cr/C and Ni-Cr/C systems with an aqueous solution of $H_2PtCl_6$ $6H_2O$ (chemical grade purity; Khimmed, Russia); then, dried and calcined at a temperature of 350 °C in a flow of $N_2$ (99.9%, 50 mL/min) for 2 h.

### 3.2. Characterization of Catalysts

The surface morphology of the catalysts was studied by transmission electron microscopy (TEM) using a JEOL-2100F (Tokyo, Japan) electron microscope in light and dark field modes at an accelerating voltage of 200 kV. The charge state of metals and the composition of the catalyst surface were determined using X-ray photoelectron spectroscopy (XPS), with a Kratos Axis Ultra DLD device (Manchester, UK) with a monochromatic radiation Al Ka (hυ = 1486.6 eV, 150 W). The standard energy of the analyzer was 160 eV and 40 eV for high-resolution spectra.

The phase composition of the samples was studied by X-ray phase analysis (XRD) using an automatic diffractometer DRON-3 (Moscow, Russia), with Cu Ka radiation (λ = 1.5405 A) with a graphite monochromator in step-by-step scanning mode (step 0.1°, exposure per point—5 s). The main 2Θ interval was 20°–90°.

The dependence of the hydrogen absorption rate in the mode of thermoprogrammable reduction (TPR) of catalysts was recorded at the laboratory installation KL-1 (Moscow, Russia). The reduction was carried out with a mixture of gases of 5% $H_2/Ar$ at a flow rate of 23 mL/min until a temperature of 850 °C. The linear heating rate of the detector was 10 °C/min.

Dehydrogenation of bicyclohexyl was carried out in a flow reactor without a carrier gas. Samples of all the studied catalysts with a weight of 1.85 g were placed in a steel reactor with an internal diameter of 10 mm and activated. Before starting the reaction, each portion of the new catalyst was precalcined in an inert atmosphere and then, reduced in a hydrogen flow. The substrate was fed at a rate of 6 mL/h (normal conditions, ρ = 0.864 $g/cm^3$) by a high-pressure pump HPP 5001. Commercial bicyclohexyl (99%, Acros Organics, Geel, Belgium, $C_{12}H_{22}$) was used as a substrate. Dehydrogenation was carried out at a temperature of 320 °C and atmospheric pressure for 4 h. Hydrogen and reaction products were separated. The hydrogen released during dehydrogenation was measured using a gas burette. To prevent the entrainment of organic substrates with a hydrogen flow, a system of cooling traps and membrane was used.

The reaction products were analyzed with a Crystallux 4000M chromatograph (Kazan, Russia) using a ZB-5 capillary column (Zebron, Phenomenex, Torrance, CA, USA) and a flame ionization detector of the FOCUS DSQ II chromato-mass spectrometer (Thermo Fisher Scientific, Waltham, MA, USA), with a TR-5ms capillary column. The analysis was performed in a programmable temperature mode of 70–220 °C at a heating rate of

6 °C/min. The purity of the hydrogen released was determined by gas chromatography with a thermal conductivity detector and a Porapak Q packed column. The conversion (X) of bicyclohexyl was calculated as the ratio of the change in the amount of bicyclohexyl before and after the reaction to the initial amount of bicyclohexyl. The selectivity (S) of the reaction products was determined as the ratio of the amount of one of the reaction products formed to the total amount.

## 4. Conclusions

Thus, the study of the composition and structure of the surface phases of mono- and bimetallic catalysts based on Ni and Cr deposited on a carbon carrier by methods of physico-chemical analysis showed the interaction of metals with each other. Due to the significant difference in the enthalpy of formation of nickel and chromium carbides when interacting with the carbon carrier, Cr atoms, when added to a bimetallic system, interact more actively with Ni than with the carrier. The data of XRD, TEM and electron diffraction indicate a change in the parameters of the nickel crystal lattice in bimetallic NiCr systems, compared with the FCC Ni lattice, which indicates, in turn, the formation of Cr-Ni solid solutions formed by the type of substitution of nickel atoms for chromium. However, this does not lead to a change in the conversion and selectivity in the bicyclohexyl dehydrogenation reaction on bimetallic NiCr catalysts, compared with the corresponding monometallic catalysts. At the same time, this circumstance leaves open the question of the high activity of trimetallic Pt catalysts deposited on the studied NiCr/C system [21,26]. Thus, the mechanism of increasing the activity of three metallic Pt catalysts deposited on the Ni-Cr/C system under study deserves additional study.

**Supplementary Materials:** The following supporting information can be downloaded at: https://www.mdpi.com/article/10.3390/catal12121506/s1, Figure S1: TEM pictures of the surface of catalysts (a) 1.5Cr/3Ni/$C_{ox}$ and (b) 0.1Pt/1.5Cr/3Ni/$C_{ox}$; electron diffraction patterns of crystallites Cr-Ni (1, 2) and Pt (3) on the surface of the 0.1Pt/1.5Cr/3Ni/$C_{ox}$ catalyst.

**Author Contributions:** Conceptualization, L.M.K.; methodology, A.N.K.; investigation, A.N.K.; writing—original draft preparation, A.N.K.; writing—review and editing, L.M.K.; supervision, L.M.K.; project administration, L.M.K.; funding acquisition, L.M.K. All authors have read and agreed to the published version of the manuscript.

**Funding:** The research was financially supported by the Ministry of Science and Higher Education of the Russian Federation (grant no. 075-15-2021-591).

**Data Availability Statement:** Data are available upon request from the authors.

**Conflicts of Interest:** The authors declare no conflict of interest.

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
