# Peer review of "Effect of Cr on a Ni-Catalyst Supported on Sibunite in Bicyclohexyl Dehydrogenation in Hydrogen Storage Application"

_catalysts, doi:10.3390/catal12121506_

Round 1

Reviewer 1 Report

Introduction: More than half of the paper is devoted to DMFC alloy catalysts, etc., but the theme of this paper is transition metal-catalyzed dehydrogenation of bicyclohexyl. The authors should discuss the difference from the related research on dehydrogenation of organic hydrides over Ni-Cr-based catalysts in more detail.

 Line 107: The peaks at 852.5 eV and 852.7 eV attributed to Ni0 valence are small shoulders, and it is questionable whether this is a significant difference.

 AlKαX-ray source was used for XPS analysis, but please describe how the charge correction was performed.

 The definitions of C and % in Table 1 should be described.

 Line 185: Since the correlation between electron diffraction and XRD was discussed in manuscript in detail, the evidence of electron diffraction should be expressed by using data.

 Line 204: The Cr alone catalyst resulted in the highest conversion of bicyclohexyl and selectivity to biphenyl, but the results of Ni-Cr alloy formation and Ni carbide formation mentioned above have little relevance to the discussion.

 The activity of Pt-based catalysts should be used as a benchmark.

 Due to the lack of discussion of characterization and catalytic activity evaluation, this paper is not considered to be at the level of publication.

Author Response

We would like to thank the reviewer for the critical comments and useful advices and questions raised. We tried to do our best to improve our manuscript by taking into account all the comments. Below we give our responses to the comments. The changes in the text are highlighted in yellow.

Comments and Suggestions for Authors

Introduction: More than half of the paper is devoted to DMFC alloy catalysts, etc., but the theme of this paper is transition metal-catalyzed dehydrogenation of bicyclohexyl. The authors should discuss the difference from the related research on dehydrogenation of organic hydrides over Ni-Cr-based catalysts in more detail.

Response: Earlier we obtained good results in the bicyclohexyl dehydrogenation reaction applied to hydrogen storage systems based on trimetallic platinum-containing catalysts. The activity of platinum in these catalysts is considered by the authors in a few publications. At the same time, the behavior of nickel-chromium systems without platinum is of great interest in order to assess their effect on platinum. The use of such systems for the dehydrogenation of organic hydrides by other researchers is unknown to the authors, therefore, as a comparison in the Introduction, examples of the use of bi- and trimetallic systems in other reactions, in particular in the electrooxidation of methanol, where nickel-chromium systems are widely used, were evaluated. At the same time, the main task of the authors was to study the effect of the interaction of modifying base metals (Cr, Ni) on the work of active platinum in the bicyclohexyl dehydrogenation reaction.

  1. Line 107: The peaks at 852.5 eV and 852.7 eV attributed to Ni0 valence are small shoulders, and it is questionable whether this is a significant difference.

Response: A similar difference in binding energies was recorded by the authors for many samples studied, including CrNi/C catalysts, which allowed us to draw conclusions about the shift and metal charging in the article. It should be noted that the shift of 0.1 eV in XPS spectra is usually considered as a measurable difference. In our case, the shift is 0.2 eV and it is reproducible in many experiments for many samples of different Ni-Cr compositions.

  1. AlKαX-ray source was used for XPS analysis, but please describe how the charge correction was performed.

Response: X-ray photoelectron spectroscopy (XPS) was used to determine the surface composition of carbon carriers and catalysts, as well as to study the content and type of surface functional groups. The spectra were recorded on the Axis Ultra DLD instrument (Kratos, UK) using monochromatic Al Ka radiation (1486.6 eV).The survey XPS spectra were obtained at an analyzer transmission energy equal to 160 eV and a step of 1 eV. High-resolution spectra were recorded with an analyzer transmission energy of 20 eV with precision fixation of individual lines and a step of 0.05 eV. The spectrometer was calibrated using the Au 4f7/2 – 83.96 eV line.

  1. The definitions of C and % in Table 1 should be described.

Response: The definitions of C and % in Table 1 are given.

  1. Line 185: Since the correlation between electron diffraction and XRD was discussed in manuscript in detail, the evidence of electron diffraction should be expressed by using data.

Response: Fig. S1 in Supplementary Information shows an example of an electron diffraction image of Cr-Ni (1, 2) and platinum (3) crystals on the surface of a 0.1Pt/1.5Cr/3Ni/Cox catalyst.

  1. Line 204: The Cr alone catalyst resulted in the highest conversion of bicyclohexyl and selectivity to biphenyl, but the results of Ni-Cr alloy formation and Ni carbide formation mentioned above have little relevance to the discussion.

Response: The conversions of bicyclohexyl dehydrogenation on Ni-Cr catalysts are incomparably lower than on Pt-catalysts, as the respected reviewer points out in the next remark. At the same time, with the combined presence of three metals, the activity of platinum increases, compared with a monometallic Pt-catalyst (see our previous paper [21]). The data presented in the reviewed article show that this may be due to the formation of Ni-Cr alloys, since Cr atoms, when added to a bimetallic system, interact more actively with Ni than with a carrier to form chromium carbide. The necessary additions were made in the discussion of the results and conclusions.

  1. The activity of Pt-based catalysts should be used as a benchmark.

Response: Necessary changes were introduced in the text (see remark 5)

  1. Due to the lack of discussion of characterization and catalytic activity evaluation, this paper is not considered to be at the level of publication.

Response: The conclusions in this article are presented in a new edition, where discussion of the formation of bimetallic Ni-Cr alloys on the catalytic activity of trimetallic Pt catalysts obtained on their basis is added.

Reviewer 2 Report

 The manuscript by Alexander Kalenchuk and Leonid Kustov, titled “Effect of Cr on a Ni-catalyst supported on Sibunite in bicyclohexyl dehydrogenation in hydrogen storage application” reports on the chemical-physical characterization and on the study of the activity of mono- and bimetallic Cr-Ni/C catalysts deposited on Sibunite as carbon carrier. TEM, EDX, XPS and XRD analyses were carried out on the prepared catalysts, which were tested in the bicyclohexyl dehydrogenation reaction.

Although in the bimetallic systems all data indicated the formation of Cr-Ni solid solutions due to substitution of nickel atoms for chromium, the activity and selectivity of bimetallic CrNi catalysts were the same of Ni and Cr monometallic ones for the bicyclohexyl dehydrogenation reaction, taken as the model reaction.

Even if the catalytic activity of the reported systems was low, the manuscript could be of interest for the readers of Catalysts journal.

I therefore suggest publication after major revision.

Althought the catalytic activity of the reported catalysts was poor compared to trimetallic PtNiCr/C system, the authors should study the recyclability of the Cr-Ni catalysts, as well as of the monometallic ones to see if activation or deactivation effects could be noticed.

Some minor points:

in table 2 specify that S stands for selectivity (it was specified several paragraphs later);

convert “non-oxidized nickel (Nio)” into “non-oxidized Ni(0)” through out the text.

Author Response

We would like to thank the reviewer for the critical comments and useful advices and questions raised. We tried to do our best to improve our manuscript by taking into account all the comments. Below we give our responses to the comments. The changes in the text are highlighted in yellow.

Comments and Suggestions for Authors

The manuscript by Alexander Kalenchuk and Leonid Kustov, titled “Effect of Cr on a Ni-catalyst supported on Sibunite in bicyclohexyl dehydrogenation in hydrogen storage application” reports on the chemical-physical characterization and on the study of the activity of mono- and bimetallic Cr-Ni/C catalysts deposited on Sibunite as carbon carrier. TEM, EDX, XPS and XRD analyses were carried out on the prepared catalysts, which were tested in the bicyclohexyl dehydrogenation reaction.

Although in the bimetallic systems all data indicated the formation of Cr-Ni solid solutions due to substitution of nickel atoms for chromium, the activity and selectivity of bimetallic CrNi catalysts were the same of Ni and Cr monometallic ones for the bicyclohexyl dehydrogenation reaction, taken as the model reaction.

Even if the catalytic activity of the reported systems was low, the manuscript could be of interest for the readers of Catalysts journal. I therefore suggest publication after major revision.

Response: The low conversion values achieved in this work indicate that Cr-Ni solid solutions themselves are not very active in the bicyclohexyl dehydrogenation reaction, which is also a very interesting and not expected result, but the formation of such a solid solution may have a stimulating effect on platinum. This result shows the way for further research and improvements in this direction, which is associated with the study of the mechanism of mutual interaction in Cr-Ni systems and the interaction of the Cr-Ni systems with platinum.

Although the catalytic activity of the reported catalysts was poor compared to trimetallic PtNiCr/C system, the authors should study the recyclability of the Cr-Ni catalysts, as well as of the monometallic ones to see if activation or deactivation effects could be noticed.

Response: The dehydrogenation process is carried out in the flow mode, therefore, the term “recyclability” is not suitable in this case to describe the performance of the catalytic systems. It would be better to talk about a long-term stability in the plug-flow runs. Both monometallic and bimetallic catalysts demonstrate a stable rate of hydrogen evolution and selectivities to the semi-hydrogenated and fully dehydrogenated products indicated in Table 2 without any noticeable loss for at least 8 hours of operation. These systems are still under study, and will be certainly studied in longer tests, but when the optimal composition of catalysts will be found.

Some minor points:

in table 2 specify that S stands for selectivity (it was specified several paragraphs later);

convert “non-oxidized nickel (Nio)” into “non-oxidized Ni(0)” through out the text.

Response: The required modifications have been made.

Reviewer 3 Report

Kalenchuk and Kustov have reported the effect of Cr on a Ni-catalyst supported on sibunite for the dehydrogenation of bicyclohexyl in hydrogen storage application. The topic is interesting, and the results are adequate. However, before the publication needs revision. The comments are the following:

1. In Fig.1, the (a) and (b) are not marked in the figure.

2. In Line 67 of Page 2, the authors mentionedAt the same time, large particles belong to nickel, and smaller ones are formed by chromium”, but this conclusion cannot be drawn from Figure 1, which lacks necessary evidence. So the author needs to give an explanation.

3. In Line 59-60 of Page 2, the authors mentioned that the catalyst contained 3 wt. % Ni and 1.5 wt. % Cr, but The EDX diagram in Fig. 1 showed that nickel and chromium seemed to be much larger than these values, please explain. In addition, the actual nickel content of Ni and Cr should be given in the manuscript, such as, determined by ICP measurement.

4. The “Ni+2” in Fig. 2 should be “Ni2+”.

5. In Line 207 of Page 6, the authors mentionedIn the case of nickel, the formation of trace amounts of cracking products [24] was also observed”. However, the total selectivity of C12H22 and C12H16 in Table 2 is still 100%, which seems contradictory. So the author needs to give an explanation. In addition, what about the carbon balance of the reaction process?

6. In Table 2, the rates of hydrogen evolution for the three bimetallic catalysts (3Ni/1.5Cr/C, (3Ni-1.5Cr)/C and 1.5Cr/3Ni/C) were all lower than that of single metal catalysts (3Ni/C and 1.5Cr/C). So what is the significance of bimetallic doping?

7. In Line 249 of Page 7, the authors mentioned “until a temperature of 700°C.” However, the maximum temperature of TPR diagram in Fig. 3 is higher than 800°C. The author needs to give an explanation.

8. Is carrier gas (hydrogen or nitrogen) required for catalytic reaction process? Does the catalyst need reduction or other pretreatment before the catalytic reaction starts? How is the Rate of hydrogen evolution calculated? These important information should be added to “3.2. Characterization of catalysts”.

9. As shown in the figure in the whole article, the catalyst sample chosen seems to be too random, such as in Fig 1. The (3Ni-1.5Cr)/C sample was chosen as the tested catalyst,but in Fig. 2. for (3Ni-3Cr)/C catalyst tested,while in Fig. 3. for catalysts (3Ni-1.5Cr)/C and 1.5Cr/3Ni/C. Why did the author design the experiments in this way? It is strongly recommended that the authors give the characterization results of all catalysts.

10. For dehydrogenation reaction, deactivation of catalyst (mainly caused by carbon deposition) is an important problem. So, what is the life of doped NiCr/C catalyst? How about carbon deposition?

Author Response

We would like to thank the reviewer for the critical comments and useful advices and questions raised. We tried to do our best to improve our manuscript by taking into account all the comments. Below we give our responses to the comments. The changes in the text are highlighted in yellow.

Comments and Suggestions for Authors

Kalenchuk and Kustov have reported the effect of Cr on a Ni-catalyst supported on sibunite for the dehydrogenation of bicyclohexyl in hydrogen storage application. The topic is interesting, and the results are adequate. However, before the publication needs revision. The comments are the following:

  1. In Fig.1, the (a) and (b) are not marked in the figure.

Response: Fig. 1 has been corrected by indicating (a) and (b) for the (3Ni-1.5Cr)/C catalyst

  1. In Line 67 of Page 2, the authors mentioned “At the same time, large particles belong to nickel, and smaller ones are formed by chromium”, but this conclusion cannot be drawn from Figure 1, which lacks necessary evidence. So the author needs to give an explanation.

Response: This sentence was modified as follows: At the same time, comparison of micrographs of TEM of a number of bimetallic Ni-Cr/C catalysts with their monometallic analogues showed that large particles be-long to nickel, and smaller ones are formed by chromium.

  1. In Line 59-60 of Page 2, the authors mentioned that the catalyst contained 3 wt. % Ni and 1.5 wt. % Cr, but The EDX diagram in Fig. 1 showed that nickel and chromium seemed to be much larger than these values, please explain. In addition, the actual nickel content of Ni and Cr should be given in the manuscript, such as, determined by ICP measurement.

Response: The catalysts were prepared by incipient wetness impregnation of the carrier, therefore, the metal contents determined by chemical analysis were exactly the same as the figures for the expected contents calculated from the concentrations of the impregnating solutions and the moisture capacity of the carrier, i.e. 3 wt. % Ni and 1.5 wt. % Cr with the maximal relative error of 0.1-0.2 wt. %.

  1. The “Ni+2” in Fig. 2 should be “Ni2+”.

Response: Fig. 2 was corrected

  1. In Line 207 of Page 6, the authors mentioned “In the case of nickel, the formation of trace amounts of cracking products [24] was also observed”. However, the total selectivity of C12H22 and C12H16 in Table 2 is still 100%, which seems contradictory. So the author needs to give an explanation. In addition, what about the carbon balance of the reaction process?

Response: Since the content of cracking products was ranged from 50 to 200 ppm, their contribution was not taken into account when calculating the selectivity of the main reaction products. This sentence was added.

  1. In Table 2, the rates of hydrogen evolution for the three bimetallic catalysts (3Ni/1.5Cr/C, (3Ni-1.5Cr)/C and 1.5Cr/3Ni/C) were all lower than that of single metal catalysts (3Ni/C and 1.5Cr/C). So what is the significance of bimetallic doping?

Response: With regard to hydrogen storage systems, the development of catalysts is limited by the need to carry out conjugated hydrogenation-dehydrogenation reactions without the formation of by-products of hydrogenolysis and cracking, which destroy the substrate and reduce the lifetime of hydrogen storage and release systems as a whole. This limits the choice of carriers and active metals, therefore, for the purposes of hydrogen storage, the study of the stimulating effect of promoting additives, in this case base metals (Cr, Ni), on the activity of Pt catalysts is of great interest. The results obtained in the work indicate that the formation of Cr-Ni solid solutions by itself does not affect the dehydrogenation of bicyclohexyl, but points to the path of further research and improvement, which is associated with the study of the mechanism of mutual interaction in Cr-Ni systems and the interaction of the Cr-Ni systems with platinum.

  1. In Line 249 of Page 7, the authors mentioned “until a temperature of 700°C.” However, the maximum temperature of TPR diagram in Fig. 3 is higher than 800°C. The author needs to give an explanation.

Response: The temperature limit in the text was corrected.

  1. Is carrier gas (hydrogen or nitrogen) required for catalytic reaction process? Does the catalyst need reduction or other pretreatment before the catalytic reaction starts? How is the Rate of hydrogen evolution calculated? These important information should be added to “3.2. Characterization of catalysts”.

Response: Dehydrogenation of bicyclohexyl was carried out in a flow reactor without a carrier gas. Before starting the reaction, each portion of the new catalyst was precalcined in an inert atmosphere, and then reduced in a hydrogen flow. The hydrogen released during dehydrogenation was measured using a gas burette. This information has been added to the section "3.2. Characterization of catalysts".

  1. As shown in the figure in the whole article, the catalyst sample chosen seems to be too random, such as in Fig 1. The (3Ni-1.5Cr)/C sample was chosen as the tested catalyst,but in Fig. 2. for (3Ni-3Cr)/C catalyst tested,while in Fig. 3. for catalysts (3Ni-1.5Cr)/C and 1.5Cr/3Ni/C. Why did the author design the experiments in this way? It is strongly recommended that the authors give the characterization results of all catalysts.

Response: The main active metal in the dehydrogenation of bicyclohexyl is platinum. Nickel and chromium are modifying components, with the combined presence of which the activity of platinum in this reaction increased [25]. The main objective of the study was to find the causes of this increase, for which the authors conducted a screening of Ni-Cr systems prepared in different ways. The article presents examples of catalysts on which the authors observed effects that could affect platinum after its introduction into the studied systems. At the same time, we regret to note that we did not have the opportunity to carry out a complete characterization of all intermediate catalysts.

  1. For dehydrogenation reaction, deactivation of catalyst (mainly caused by carbon deposition) is an important problem. So, what is the life of doped NiCr/C catalyst? How about carbon deposition?

Response: Under optimal conditions of the bicyclohexyl dehydrogenation reaction, we did not observe side reactions of cracking and hydrogenolysis, as well as any deactivation of the catalyst. The doped NiCr/C and monometallic catalysts demonstrated a very stable activity for at least 10 h.

Round 2

Reviewer 1 Report

The authors have revised thes paper appropriately to a certain degree, and I consider it to be above the level for acceptance.

Reviewer 2 Report

The authors adequately responded to all reviewers' suggestions and concerns, therefore I suggest publication in Catalysts journal .

Reviewer 3 Report

The authors worked on the comments. Therefore, the revised manuscript can be accepted.